# OpenReview forum: "A Variational Framework for Improving Naturalness in Generative Spoken Language Models"
_ICML.cc/2025/Conference — ICML 2025 poster_

### Official Review · Reviewer_odWp · 2025-03-09

**Overall Recommendation:** 4

**Summary:**

This paper proposes a variational approach to speech-language modelling in contrast to traditional auto-regressive models. The aim is to capture information other than semantics.

## update after rebuttal
I checked the results in Appendix H and I think the results are interesting. Why not add those results into the main paper (since we as reviewers are not required to see Appendices)? I increase my score to 4 to acknowledge this effort.

**Claims And Evidence:**

Yes. I can find evidence in section 5 supporting the authors claim that the variational method improves the naturalness of synthesised speech.

**Essential References Not Discussed:**

G. Sun et al. "Generating diverse and natural text-to-speech samples using a quantized fine-grained vae and autoregressive prosody prior", In Proc. ICASSP. 2020.

This should be discussed in section 3.1. This paper is the first to apply a trainable auto-regressive prior in speech synthesis.

**Experimental Designs Or Analyses:**

The experimental design is sound and valid. My main concern is how this method is useful for more practical downstream tasks such as ASR, emotion or speaker recognition, to reflect that capturing the additional (mainly paralinguistic) information is useful. I strongly encourage the authors to conduct at least 2 of the above practical tasks using the variational speech LM and compare it to token-based speech LM to see any potential benefits of using variational methods.

**Methods And Evaluation Criteria:**

Evaluation is comprehensive.

**Other Comments Or Suggestions:**

Actually, Fig. 1 can be improved by unifying the font.

**Other Strengths And Weaknesses:**

The experiment was conducted using LibriSpeech and Libri-light, which are datasets with quite small variabilities other than semantic information. I believe the variability remains in the speaker representation space, which is not explicitly reflected in the experimental design.

**Questions For Authors:**

N/A

**Relation To Broader Scientific Literature:**

This is to my best knowledge the first work to incorporate variational framework in speech language models.

**Theoretical Claims:**

Not applicable.

---

> ### Author Rebuttal · Authors · 2025-04-01
>
> We sincerely thank the reviewer for recognizing both the robustness of our experimental design and the novelty of our contribution. Below, we address each concern raised:
>
> ## Emotion and Speaker Recognition
>
> > My main concern is how this method is useful for more practical downstream tasks such as ASR, emotion or speaker recognition, to reflect that capturing the additional (mainly paralinguistic) information is useful. I strongly encourage the authors to conduct at least 2 of the above practical tasks using the variational speech LM and compare it to token-based speech LM to see any potential benefits of using variational methods.
>
> We appreciate this valuable feedback. We would like to direct the reviewer to Appendix H, where we present our comprehensive evaluation results on speech emotion recognition and speaker recognition tasks. In Tables 8 and 9, our results suggest that the variational features encode speaking styles and prosodic patterns there are useful for both tasks. We kindly direct the reviewer to Appendix H for more detailed experimental setup, results and analysis. Additionally, we would like to note that our main objective remains to improve naturalness of speech synthesis, and the side experiments provide additional evidence that our method is capable of achieving that.
>
>
> ## Additional References
>
> > G. Sun et al. "Generating diverse and natural text-to-speech samples using a quantized fine-grained vae and autoregressive prosody prior", In Proc. ICASSP. 2020.
> This should be discussed in section 3.1. This paper is the first to apply a trainable auto-regressive prior in speech synthesis.
>
> Thank you for this important suggestion. We will incorporate this reference in Section 3.1 and properly acknowledge its contribution.
>
> ## Figure Presentation
>
> > Actually, Fig. 1 can be improved by unifying the font.
>
> We appreciate this attention and will unify the font in Fig. 1.
>
> ## Dataset Selection
>
> > The experiment was conducted using LibriSpeech and Libri-light, which are datasets with quite small variabilities other than semantic information. I believe the variability remains in the speaker representation space, which is not explicitly reflected in the experimental design.
>
> We thank the reviewer for their valuable feedback. We acknowledge that LibriSpeech and Libri-light are not considered highly expressive. We evaluate on these datasets since they are standard benchmarks in the literature [1, 2, 3]. Importantly, our proposed approach improves synthesis naturalness even on these less expressive datasets. This suggests that our method would likely yield even greater improvements on more expressive datasets, where natural expressive speech synthesis presents additional challenges.
>
> We hope these responses adequately address the reviewer's concerns and help in the evaluation of our work.
>
> ## References
>
> [1] E. Kharitonov, et al., Text-free prosody-aware generative spoken language modeling, in Proceedings of the 60th Annual Meeting of the Association for Computational Linguistics (Volume 1: Long Papers), pp. 8666–8681, Dublin, Ireland, May 2022.
>
> [2] Z. Borsos, et al., AudioLM: A language modeling approach to audio generation, IEEE/ACM Transactions on Audio, Speech, and Language Processing, 31:2523–2533, 2023.
>
> [3] K. Lakhotia, et al., On Generative Spoken Language Modeling from Raw Audio, Transactions of the Association for Computational Linguistics, 2021.

---

### Official Review · Reviewer_Pvan · 2025-03-12

**Overall Recommendation:** 3

**Summary:**

This paper automatically learns continuous speech attributes (such as pitch, energy, spectrum) through VAE, and jointly model them with semantic tokens to improve the naturalness and language fidelity of generated speech. Experiments show that this method significantly outperforms the baseline model in subjective naturalness scores and does not require manual design of rhythmic features.

**Claims And Evidence:**

Yes.

**Essential References Not Discussed:**

No.

**Experimental Designs Or Analyses:**

Yes.
The experimental design verifies the effectiveness of the proposed variational method by comparing the three baseline methods of Token-LM, Token-LM+Pitch and Token-LM+Acoustic, and analyzes its regulatory effect on information encoding and generation quality through hyperparameter β and γ ablation studies.

**Methods And Evaluation Criteria:**

Yes.

**Other Comments Or Suggestions:**

No.

**Other Strengths And Weaknesses:**

In this paper, a good balance is struck between innovation and practicability, and the continuous prosodic features are automatically learned by the VAE, which not only avoids the cumbersome process of manually designing prosodic features in the traditional method, but also improves the naturalness of the generated speech by jointly modeling semantic and prosodic information.

The experimental design is also rigorous, and the validity of the method is verified by multi-dimensional indicators (reconstruction quality, language modeling ability, subjective scoring), and the subjective evaluation adopts standardized processes, which enhances the credibility of the results.

But this "end-to-end" design idea does not seem to be new in the field of speech synthesis.
And the performance of the model is highly dependent on the tuning of β and γ, which may limit the robustness of the method under different data distributions, as wel as the experiment only verifies the English dataset, while the prosodic patterns of different languages are quite different, and it is unclear whether the model needs to be retrained or adjusted hyperparameters.

But overall, the paper proposes a valuable solution to the problem of naturalness of speech generation, so I give a weak acceptance.

**Questions For Authors:**

see other strengths and weakness

**Relation To Broader Scientific Literature:**

This paper continues the idea of [1] to improve the naturalness of generation by combining modeling language and prosodic information, but abandons the hand-designed fundamental frequency features and moves towards end-to-end learning.

In contrast to the discrete acoustic token method of [2], the advantages of continuous latent variables in prosodic modeling are verified, and the information loss caused by multi-stage discretization is avoided.

[1] Kharitonov, E., Lee, A., Polyak, A., Adi, Y., Copet, J., Lakhotia, K., Nguyen, T. A., Riviere, M., Mohamed, A., Dupoux, E., and Hsu, W.-N. Text-free prosody-aware generative spoken language modeling. In Proceedings of the 60th Annual Meeting of the Association for Computational Linguistics (Volume 1: Long Papers), pp. 8666– 8681, Dublin, Ireland, May 2022. Association for Computational Linguistics. doi: 10.18653/v1/2022.acl-long. 593.

[2] Borsos, Z., Marinier, R., Vincent, D., Kharitonov, E., Pietquin, O., Sharifi, M., Roblek, D., Teboul, O., Grangier, D., Tagliasacchi, M., and Zeghidour, N. Audiolm: A language modeling approach to audio generation.

**Theoretical Claims:**

Yes.
The ELBO decomposition and KL divergence derivation are mainly checked to comply with the standard theory of variational autoencoder and are correct.

---

> ### Author Rebuttal · Authors · 2025-04-01
>
> We sincerely thank the reviewer for the thorough review and acknowledging the strengths of our paper, including the balance between innovation and practicability, the automatic learning of continuous prosodic features, and our rigorous experimental design.
>
> Below, we address the feedback from the reviewer:
>
> ## End-to-End Design Novelty
>
> > This "end-to-end" design idea does not seem to be new in the field of speech synthesis.
>
> We thank the reviewer for this observation. End-to-end is indeed the current trend for building speech synthesis models. Our specific contribution lies in making the prosodic features end-to-end learnable with the speech language model through the integration with VAE. This approach enables automatic learning of continuous prosodic features without manual feature engineering, which we believe represents a meaningful advancement in the field.
>
> ## Hyperparameter Sensitivity
>
> > The performance of the model is highly dependent on the tuning of β and γ, which may limit the robustness of the method under different data distributions.
>
> This is an insightful observation. We agree that hyperparameter sensitivity is an important consideration for most work in this field. In Section 8: Limitations and Future Work, we have acknowledged this challenge and identified exploring automated methods for hyperparameter tuning as an important direction for future research. The current work serves as a first and crucial step in demonstrating that the proposed method effectively improves the prosodic naturalness of speech language models.
>
> ## Cross-Lingual Adaptability
>
> > The experiment only verifies the English dataset, while the prosodic patterns of different languages are quite different, and it is unclear whether the model needs to be retrained or adjusted hyperparameters.
>
> This is an excellent point that highlights an important direction for future research. We hypothesize that our approach of learning continuous prosodic features automatically rather than using hand-designed features offers inherent advantages for cross-lingual adaptation. Unlike rule-based systems that require language-specific expertise, our VAE framework can potentially discover and model language-specific prosodic patterns without human supervision.
>
> We will add this discussion to the limitations section and highlight it as a direction for future work. Our approach, which learns continuous representations rather than relying on predefined features, may be well-suited to adapt to the diverse prosodic patterns across languages, though this remains to be verified through additional research.
>
>
> We thank the reviewer for the valuable suggestions, which will help direct our ongoing research efforts.

---

> > ### Comment · Reviewer_Pvan · 2025-04-04
> >
> > Thanks for the detailed reply. I acknowledge the contributions of this study. However, the author's reply is not enough to dispel my concerns about the weaknesses mentioned above. Therefore, I tend to keep my original score.

---

> > > ### Author Response · Authors · 2025-04-08
> > >
> > > Thank you for your thoughtful consideration of our rebuttal and for your active response. We sincerely appreciate your acknowledgment of our contribution to the literature as well as the feedback that helps us refine our work.

---

### Official Review · Reviewer_XtBM · 2025-03-13

**Overall Recommendation:** 3

**Summary:**

This paper proposes a variational framework to enhance the naturalness of generative spoken language models by jointly learning continuous paralinguistic features and discrete semantic tokens. Traditional token-based models often neglect prosodic information, leading to unnatural speech. The authors address this by integrating a variational autoencoder (VAE) with an autoregressive prior to automatically encode continuous speech attributes (e.g., pitch, energy) alongside pre-extracted semantic tokens.

**Claims And Evidence:**

Yes.

**Essential References Not Discussed:**

No.

**Experimental Designs Or Analyses:**

The comparison between the proposed and baseline methods is not sufficient. Generally, this work models the residual acoustic features to minimize the information loss of semantic token based LLM generation.

Generally, there is some existing (not limited) serial of works: 1)  some works such as SpeechTokenizer [1] directly distill semantic tokens in the first layer of RVQ and use AR to generate the semantic tokens and NAR model to predict the rest. This is different from the setting in "Token-LM acoustic" since the acoustic tokens are learned without semantic token supervision, and the generative process is not well-designed. 2) some works only use AR/NAR methods to generate the semantic tokens and model the acoustic features by an additional diffusion model. It can also model the paralinguistic features to enhance the naturalness. These works include but not limited to MaskGCT[2], CosyVoice[3], tortoise-tts[4]

[1] SpeechTokenizer: Unified Speech Tokenizer for Speech Large Language Models
[2] MaskGCT: Zero-Shot Text-to-Speech with Masked Generative Codec Transformer
[3] Cosyvoice: A scalable multilingual zero-shot text-to-speech synthesizer based on supervised semantic tokens
[4] tortoise-tts: Better speech synthesis through scaling.

My questions are:
* 1. The author should compare with more baselines, such as SpeechTokenizer, MaskGCT, et. al.
* 2. The author should discuss more with existing works. why such VAE is necessary? What benefits can it bring compared with existing works?
* 3. The "Token-LM acoustic" shows good reconstruction results but weak generation performance. But I think it comes from: 1) the tokenizer used is not reasonable, at least, the author should compare it with SpeechTokenizer; 2) the generative model is also not well designed. I suggest using AR to predict the first semantic tokens and NAR to predict the residual acoustic features which use semantic tokens as conditions.

**Methods And Evaluation Criteria:**

The method is well-written and easy to follow.

Some questions:

1.  This paper proposes VAE with autoregressive prior, to learning acoustic features to enhance naturalness in semantic token based LLM. However, some work such as SpeechTokenizer which uses RVQ-based methods and distills semantic tokens in the first layer can also address the issues: present the paralinguistic features while modeling the semantic features. The author should discuss them.

[1] SpeechTokenizer: Unified Speech Tokenizer for Speech Large Language Models

2. Why use CER instead of WER? The speaker similarity metric is not reported.

**Other Comments Or Suggestions:**

No.

**Other Strengths And Weaknesses:**

The strengths and weakness are discussed in the previous sections.

**Questions For Authors:**

The questions are discussed in the previous sections.

**Relation To Broader Scientific Literature:**

No.

**Theoretical Claims:**

Yes.

---

> ### Author Rebuttal · Authors · 2025-04-01
>
> We sincerely thank the reviewer for their constructive feedback. We address your concerns comprehensively below. We present the Tables of new experiment results [here](https://anonymous.4open.science/api/repo/icml-rebuttal-BDD8/file/ICML-rebuttal-1.pdf).
>
>
>
>
> ## Comparison with Additional Baselines
> > Some work such as SpeechTokenizer which uses RVQ-based methods and distills semantic tokens in the first layer can also address the issues: present the paralinguistic features while modeling the semantic features. The author should discuss them.
>
>
> > The author should compare it with SpeechTokenizer; 2) the generative model is also not well designed. I suggest using AR to predict the first semantic tokens and NAR to predict the residual acoustic features which use semantic tokens as conditions.
>
>
> We appreciate your points about comparing our approach with additional baselines. We acknowledge that our comparison could be more comprehensive and have implemented a SpeechTokenizer-based approach as suggested.
>
>
> **SpeechTokenizer Baseline**: We used the official SpeechTokenizer checkpoint for the encoder and decoder components (semantic and acoustic tokens ↔ speech). But we trained our own NAR decoder for predicting acoustic tokens from semantic tokens, as the original implementation requires text input (it was originally designed for TTS). We also re-trained the autoregressive model for semantic token prediction due to token set differences (1024 vs 200 tokens).
>
>
>
>
> As shown in [Table 2](https://anonymous.4open.science/api/repo/icml-rebuttal-BDD8/file/ICML-rebuttal-1.pdf), our approach still achieves superior naturalness and meaningfulness scores compared to the new established SpeechTokenizer baseline. Perceptual listening revealed that the SpeechTokenizer approach indeed improves upon Token-LM in prosody patterns. However, it introduces more audio quality artifacts compared to Token-LM which uses a diffusion-based decoder to convert semantic tokens to speech. These artifacts offset the benefits of better prosody naturalness in the human evaluation scores. Note that we re-evaluate all methods in the human evaluation for fair comparison.
>
>
> We will revise Sections 5 and 6 to incorporate these new results.
>
>
> ## Roles of VAE and Comparison with Existing Works
>
>
> > Why such VAE is necessary? What benefits can it bring compared with existing works?
>
>
> Our approach offers distinct advantages over existing methods:
> 1. **Compared to SpeechTokenizer and Token-LM**: In these baselines, the AR model accesses only semantic tokens. Our AR model additionally accesses variational features encoding prosody, better utilizing the stronger AR model for improved naturalness.
> 2. **Compared to Token-LM+Acoustic**: The acoustic (RVQ) tokens from this baseline is extracted without supervision from the AR model, while our approach jointly optimizes variational features for both reconstruction and AR prediction, where the encoder receives training signal from the AR model.
> 3. **Compared to Token-LM+Pitch**: Our approach doesn't require hand-engineered features, as it learns prosodic information in an unsupervised fashion.
>
>
> The VAE framework enables joint learning of variational features optimized for both reconstruction and AR prediction, yielding superior naturalness and meaningfulness of the syntheses. We'll revise Section 6 to emphasize these advantages.
>
>
>
>
> ## Additional Metrics
>
>
> > Why use CER instead of WER?
>
>
> We originally chose CER following prior work [1], which evaluates pronunciation errors more accurately. As requested, we've added WER measurements in [Table 1](https://anonymous.4open.science/api/repo/icml-rebuttal-BDD8/file/ICML-rebuttal-1.pdf). The WER results follow the same trend as CER.
>
>
> > The speaker similarity metric is not reported.
>
>
> In [Table 1](https://anonymous.4open.science/api/repo/icml-rebuttal-BDD8/file/ICML-rebuttal-1.pdf), we add speaker similarity metrics, which is calculated using the cosine similarity of the speaker embeddings extracted from a pre-trained speaker verification model. Our method achieves a speaker similarity score slightly lower compared to the baseline approaches (0.42 v.s. 0.45). However, as the reviewer can verify from our audio samples, the perceptual difference in speaker identity is minimal, while naturalness and intelligibility improvements are noticeable. This represents a reasonable trade-off for most speech applications.
>
>
> We believe these additions and clarifications address the reviewer's concerns and strengthen our paper.
>
>
> ## References
>
>
> [1] K. Lakhotia, et al., On Generative Spoken Language Modeling from Raw Audio, Transactions of the Association for Computational Linguistics, 2021.

---

### Official Review · Reviewer_enQh · 2025-03-14

**Overall Recommendation:** 1

**Summary:**

The authors propose a variational approach that directly encodes desired information from raw audio inputs, addressing the challenge of preserving prosody when modeling discrete speech codes primarily focused on phonetic content (such as HuBERT tokens). Their variational approach shows improved performance compared to baselines such as Token-LM (phoneme-based tokens only), Token-LM+acoustic (phoneme and acoustic tokens), and Token-LM+acoustic baseline.

**Claims And Evidence:**

The comparisons in this work mainly focus on Token-LM models derived from self-supervised models like HuBERT, which do not use a reconstruction loss. However, if the model were compared to more recent approaches using more compressed acoustic tokens such as WavTokenizer, it would likely provide better insight into prosody than an LLM trained only on SSL-based semantic tokens. This weakens the motivation behind the proposed approach.

**Essential References Not Discussed:**

.

**Experimental Designs Or Analyses:**

In terms of experimental design, the baselines used are re-implemented versions of existing models rather than direct comparisons with the original models. As a result, the validity of the reported evaluation results is somewhat diminished.

**Methods And Evaluation Criteria:**

Not really. The use of a VAE to encode information absent in SSL-based discrete semantic tokens is reasonable. However, the model employs a diffusion decoder to reconstruct continuous representations, requiring 100 iterative steps for generation. This significantly increases computational complexity compared to predicting discrete acoustic codecs, making it unsuitable for real-time speech generation applications.

**Other Comments Or Suggestions:**

.

**Other Strengths And Weaknesses:**

Weakneeses

The authors compare their approach only with re-implemented models rather than conducting direct comparisons with existing baseline models. Since the baseline is based on Token-LM, which models very small discrete tokens with only 200 clusters, the results are not sufficiently comprehensive.

Methodologically, as mentioned above, it is difficult to determine what advantages this approach has over LLM-based models that predict acoustic tokens with delay patterns. Additionally, the reliance on a diffusion decoder raises concerns about whether this method can be used in streaming scenarios.

**Questions For Authors:**

What is the real-time factor (RTF) of the model's generation speed?

Can the mel-spectrogram be decoded only after all continuous features have been generated?

**Relation To Broader Scientific Literature:**

While this approach could serve as an alternative to acoustic codecs, its reliance on a separate diffusion model for continuous feature generation increases computational cost significantly. Consequently, it is unlikely to be practical for real-time generation or streaming scenarios.

**Theoretical Claims:**

The VAE formulation is correct, and I have reviewed the loss derivation provided in the appendix.

---

> ### Author Rebuttal · Authors · 2025-04-01
>
> We appreciate both the recognition of our formulation's correctness and the critical feedback from the reviewer. We address the feedback below.
>
> ## Computational Complexity
> >The model relies on a diffusion decoder, which increases computational complexity compared to predicting discrete acoustic codecs, making it unsuitable for real-time speech generation applications.
>
> We would like to clarify several points:
>  - **Decoder flexibility**: Our variational approach isn't tied to a specific decoder. We chose diffusion for easier training, but our method works with various decoding strategies, including discrete token prediction. There is no constraint on the decoder $\theta$ used to parameterize $p_\theta(\mathbf{X}\mid\mathbf{Z})$. To validate this, we conducted additional experiments replacing the diffusion decoder with a token decoder which converts semantic tokens to SpeechTokenizer tokens [1], then used a pre-trained SpeechTokenizer decoder to generate speech. [Table 1](https://anonymous.4open.science/api/repo/icml-rebuttal-BDD8/file/ICML-rebuttal-2.pdf) shows that this variant (*Proposed + Token Decoder*) achieves similar or better performance compared to the diffusion approach.
>  - **Fair comparison**: We used the same diffusion decoder architecture across all comparing methods. We ensure a fair comparison where only the modeling approach varies, isolating the impact of our variational method.
>  - **Research focus**: Our primary contribution is on improving prosodic naturalness with the variational approach, not on optimizing for real-time applications. The improvement in naturalness MOS validates the effectiveness of our method.
>  - **Reasonable RTF**: On a single NVIDIA L40S GPU with batch size 1, our current implementation achieves an RTF of 0.68 (<1). While this is not our focus, it shows that this approach remains practical. Importantly, the diffusion decoder only operates once after the AR model completes its generation, so the additional latency introduced is not as significant as it might initially appear.
>
> We will make adjustments to Sections 3 and 4 to emphasize these points and avoid confusion.
> >Can the mel-spectrogram be decoded only after all continuous features have been generated?
>
> Yes. Our current implementation with diffusion decoder and HiFi-GAN vocoder is not streamable. However, as mentioned above, our framework is compatible with any type of decoder, including streamable ones.
>
> ## Use of Re-implemented Models
> >The baselines used are re-implemented versions of existing models rather than direct comparisons with the original models.
>
> We re-implemented the baseline models following standard scientific methodology to ensure a controlled and fair comparison. This decision was made for several critical reasons:
>
> - **Architectural consistency**: We needed to maintain consistent architecture and model size across all methods. Our re-implementations use identical decoder and autoregressive model architectures, differing only in the specific modeling approach being evaluated. This isolates the impact of our variational method, which is the primary contribution of our work.
> - **Data consistency**: We ensured all models were trained on exactly the same data. Off-the-shelf implementations typically differ in training datasets, preprocessing pipelines, and model sizes, introducing confounding variables that would make it impossible to attribute performance differences specifically to our methodological innovation.
>
> ## Comparison with More Recent Approaches
> >If the model were compared to more recent approaches using more compressed acoustic tokens such as WavTokenizer, it would likely provide better insight into prosody than an LLM trained only on SSL-based semantic tokens.
>
> We include the Token-LM + Acoustic baseline, which employs acoustic tokens from residual vector quantization (RVQ), capturing information beyond what semantic tokens.
> To compare with recent approaches on acoustic tokenization, we have included an additional comparison with SpeechTokenizer in [Table 1](https://anonymous.4open.science/api/repo/icml-rebuttal-BDD8/file/ICML-rebuttal-2.pdf) (details in our response to Reviewer XtBM). This expanded analysis further validates the advantages of our variational method across different tokenization techniques.
>
> ## Token-LM Baseline with Limited Clusters
> >Token-LM models very small discrete tokens with only 200 clusters, the results are not sufficiently comprehensive.
>
> We specifically chose k=200 for our HuBERT baseline after conducting a sweep across different values (k=50, 200, 1000), finding that k=200 performed the best for language modeling, as detailed in Appendix F. Additionally, the new added SpeechTokenizer baseline has 1024 clusters, which our model still outperforms.
>
> Thank you for your valuable feedback which has helped us articulate our contributions more clearly.
> ## References
>
> [1] X. Zhang et al., SpeechTokenizer: Unified Speech Tokenizer for Speech Language Models, ICLR, 2024

---

### Decision · Program_Chairs · 2025-05-01

**Decision:**

Accept (poster)

**Comment:**

The authors propose a VAE approach to capture prosodic information atop semantic speech tokens, in contrast to discrete, possibly designed acoustic features (pitch like in pGSLM, or acoustic tokens like Encodec). This is a replacement with learned continuous representations (R3: the end-to-end part is not novel, and designing has traded with having to tune beta-VAE parameters; the scheme and use-case is, however). Generations from the resulting speech LMs match baseline zero-shot metrics while being more meaningful and notably more natural per subjective metrics. Though not linked in the body (I suggest the authors do), the Appendix showed improvements on emotion recognition and speaker ID, evidence of general effectiveness (led R4 to raise score to 4).

R1 raised valid concerns, such as the cost of diffusion (though authors clarify it is still better than real-time, plus streaming is not the goal). Authors note to R1 that the VAE method is not decoder-specific, and show the SpeechTokenizer decoder also works (a start to addressing R2’s request for more baselines). However, the improvements from that plus use of reimplemented models (to isolate the variational scheme), suggest their baseline may be weak.

Reviewers generally agree the method is sound and well-presented. Though R1 raised key concerns, they seem to be mitigated and the other three support acceptance, with one solid accept. The work’s results validate a novel alternate approach to current discrete acoustic tokenizers in speech generation, with promising results on three tasks. To more convincingly improve current methods, authors should operate on stronger baselines (theirs have a relatively large gap w/ ground truth MOS) and reduce the tuning effort needed (but their honesty here is appreciated). I suggest Weak Accept.

Note: R1’s final update was not visible to authors, so I share here:

```python
The authors emphasize architectural and data consistency across experiments to ensure fair comparison, demonstrating that their proposed variational approach outperforms ablated versions that use only tokens, pitch, or acoustic information. However, the paper lacks comparison against strong external spoken language model baselines that have been carefully optimized, which significantly limits the claim of superiority. This absence of external benchmarks is a notable weakness.

Additionally, to support decoder flexibility, the authors replace the diffusion-based decoder with a tokenizer-based one, showing improved results (https://anonymous.4open.science/api/repo/icml-rebuttal-BDD8/file/ICML-rebuttal-2.pdf). However, this suggests that the diffusion decoder used throughout the paper may not be optimal. As noted in the initial review, diffusion decoders are generally not ideal for streaming applications, which reduces the practical value of the current setup. A stronger decoder choice—especially one that aligns better with practical use cases like streaming—would have strengthened the paper’s overall contribution.

While the paper aims to demonstrate the value of the variational approach, the lack of external baseline comparisons and concerns around the practical utility of the chosen decoder leave the impact of the proposed method unclear. Given these limitations, I lean toward a score between Reject (1) and Weak Reject (2).
```